# Anomalous Action Recognition via Spatio-temporal Relation and Key Patch Selection

## Abstract

For providing timely warnings and preventing potential damages, it is crucial to detect anomalous actions that threaten public safety through surveillance cameras. Compared to normal actions, anomalous actions often occupy only a small portion of surveillance videos and exhibit more complex manifestations in terms of time and space. Considering that normal action recognition methods fail to highlight crucial information from small-sized patches, resulting in imprecise anomaly modeling, we propose the Spatio-Temporal Key Patch Selection Network (SKPS-Net). To tackle the challenge of detecting anomalous behaviors that manifest in small and inconspicuous areas, we design a spatial adaptive key patch selection module to select small but informative patches on input videos. Furthermore, the long-short feature map spatio-temporal relation module is devised to make the key patch effectively capture the continuous dynamic changes of anomalous actions. Finally, we propose a spatio-temporal refined loss to reinforce fine-grained feature learning. Experiments conducted on the HMDB51, Kinetics, and UCF-Crime v2 datasets demonstrate that our SKPS-Net achieves state-of-the-art performance in few-shot action recognition, outperforming the most competitive methods by 1.2% on the anomalous action dataset UCF-Crime v2.

## 1 Introduction

Anomalous actions, such as fights, arson, robbery, etc., can pose significant threats to public safety. Surveillance videos often serve as the initial source for capturing anomalies. By recognizing the anomalous actions in the video, appropriate safety measures can be swiftly formulated based on the nature of the detected activity. In this paper, an autonomous method to make anomalous action recognition is proposed.

There are many difficulties in recognizing anomalous actions. To begin with, anomalous action typically manifests in a more intense and irregular way(Zhou et al., 2019), which requires more precise features for recognition. However, many existing models only consider the global feature(the feature of the whole input frame)(Wang et al., 2021b; Perrett et al., 2021; Nguyen et al., 2022), and some of them are employed for anomalous action recognition. The objects in the anomalous action, including the person and the car, can be far from the surveillance camera and have a relatively small size, which means the discriminative information only exists in small local patches(Xiao et al., 2023b;a). These patches are called key patches due to the fact that they make more contributions to the recognition task. As a result, it is hard to model the anomalous action only using the global feature. Besides, the video contains complicated spatio-temporal changes, like continuous walking and abrupt body movements, which are also critical details for anomalous action recognition. Consequently, the selection of the key patch needs to rely on spatio-temporal information(Wang et al., 2021a; 2022b).

Considering that vital information is concentrated in the local key patches, which makes the global feature modeling of the anomalous action difficult, our work introduces spatial adaptive key patch selection to enhance global feature representation. Using the object detection network to locate the patch will highly raise the cost, so the lightweight module is proposed in this paper. Unlike some methods that locate patches using the global feature vector (Wang et al., 2021a; 2022b), we suggest that the feature map, which retains two-dimensional spatial information, is more effective

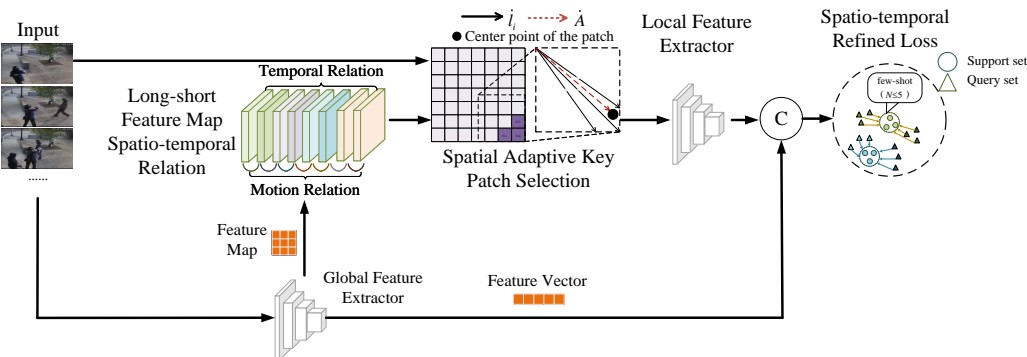

Figure 1: The architecture of SKPS-Net.

for accurate patch localization. The module in this paper takes the feature map as the input and fully mines the spatial information. This module achieves effective selection without position annotation and additional model weight, which means less training cost and plug-and-play component.

Moreover, the patch selection also need to incorporate spatio-temporal information to capture the detailed changes in the video. Whereas other methods focus on the patch-level information within the individual frame(Thatipelli et al., 2022), we propose the long-short feature map spatio-temporal relation module. Spatio-temporal modeling is usually considered at different scales(Wang et al., 2021b; Jiang et al., 2019): long-range temporal aggregation and short-range motion. The existing networks for spatio-temporal modeling, like 3D convolutional network (3D CNN) and Flownet, are hard to integrate into the framework due to their large size. We use the lightweight module based on 2D convolution consisting of two submodules to make temporal and motion relations. With more than individual frame information, the related feature map can focus more on the patch where the action happens.

After the feature enhancement, the proper loss function is required to improve the generalization ability. The simple loss function of few-shot learning mainly comes out of the final output features of the network. We propose to calculate the loss with more refined spatial and temporal features. The loss includes features from more levels and helps improve the feature robustness.

The spatio-temporal key patch selection network (SKPS-Net) for anomalous action recognition is constructed, as can be seen in Fig.1. It uses few-shot learning to solve the lack of anomalous action data. The main contributions of our paper can be summarized as follows:

(1)The plug-and-play spatial adaptive key patch selection module is proposed to highlight the vital but less obvious local information. It utilizes the spatial information within the feature map to adaptively select the key patch, modeling the patch-level anomalous action.

(2)The long-short feature map spatio-temporal relation module is proposed to enrich the information for selection in (1). It relates the long-range temporal information and short-range motion information of feature maps in the same video, enabling better localization of the key patch in videos with sophisticated spatio-temporal changes.

(3)The spatio-temporal refined loss function uses multi-level features for the recognition task, improving the generalization capability. It conforms to the enhanced feature after (2) and (1), jointly learning features from different subspaces, and improves class-specific temporal discriminability using frame-level features.

## 2 METHOD

### 2.1 NETWORK ARCHITECTURE

We propose the SKPS-Net for anomalous action recognition. While the global feature vector is widely used, we try to exploit the potentiality of the global feature map to take advantage of features

at multiple levels. In more detail, the network enriches the spatio-temporal context by connecting, highlights the core information in complex scenarios, and makes feature extraction from high-value and small-size input.

Considering the difference in quantity, the anomalous action recognition can be seen as the few-shot action recognition task. The few-shot learning gives the model task-level knowledge from the large normal action dataset, so it can avoid the expense of expanding the existing datasets which contain mostly normal action data (Li et al., 2013; Luo et al., 2017) or have insufficient quantity and variety of anomalous actions(Sultani et al., 2018; Öztürk & Can, 2021). The network learns the C-way K-shot classification task in each episode. The input of the network includes the support set comprising K labeled instances for each of the C classes and the query set of unknown samples. The goal is to classify the unlabeled video in the query set according to the features and labels of the support set. We follow the episodic training paradigm in prior works(Vinyals et al., 2016; Finn et al., 2017), and randomly sample the tasks from the training set. A sequence of uniformly sampled frames serves as the representation for each video.

As can be seen in Fig.1, we add a key patch branch on the commonly employed global feature network. The feature map and feature vector extracted are separately used for local and global representation. Then the long-short feature map spatio-temporal relation module can connect the temporal and motion relations in the video. After connection, the feature maps are fed into the spatial adaptive key patch selection module to get the small patch with distinctive objects out of the original frame. The key patch feature extraction can be completed quickly because of the small inputs. After feature enhancement, the improved features of the support set and query set calculate the spatio-temporal refined loss to match the action.

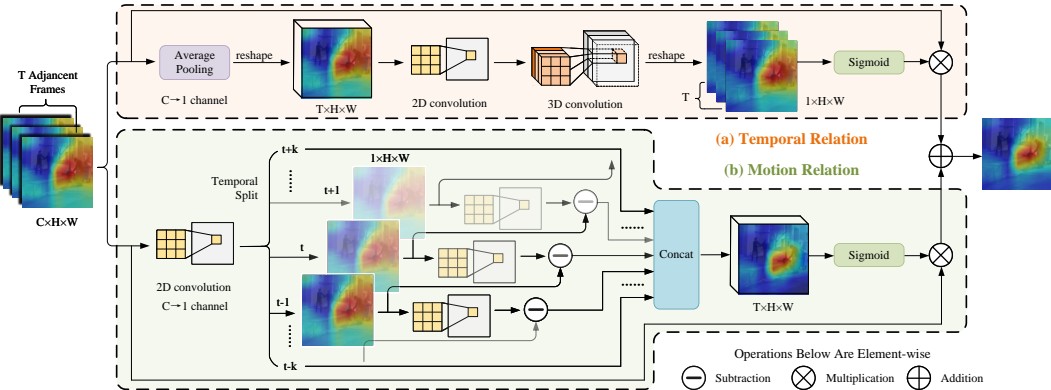

Figure 2: Illustration of long-short feature map spatio-temporal relation module including two sub-modules. (a) Temporal relation module. (b) Motion relation module. Best viewed in color and zoomed in.

## 2.2 LONG-SHORT FEATURE MAP SPATIO-TEMPORAL RELATION MODULE

The global feature map only reserves the spatial information from the individual frame and lacks the representation of the entire video information. In addition to the static spatial information, spatio-temporal changes are also crucial for effectively locating key patches in the video. More specifically, the motion information represents short-range changes between neighboring frames, highlighting areas with sudden intense changes, while temporal information, derived from stacked frames, denotes continuous long-range evolution. These two kinds of information are complementary and need collaborative utilization. Based on light-weight convolution modules(Jiang et al., 2019; Li et al., 2020), the long-short feature map spatio-temporal relation module is proposed, as shown in Fig.2. It consists of two submodules: the temporal relation module and the motion relation module. The final feature map with spatio-temporal information is created by the element-wise addition of the outputs from two submodules. The module successfully establishes the temporal and motion relation between frames at the feature map level.

### 2.2.1 TEMPORAL RELATION MODULE

Using temporal information can ensure that the key patches selected at different timestamps correspond to the long-range changes. 3D convolutions can jointly learn the spatial and temporal features from continuous frames. But the temporal information in the whole video is not mixed up at once only using 3D convolution, because of the limited kernel size and the restricted computational cost. So the 2D convolution is factorized to make a whole-range temporal relation. Then the 3D convolution can process the features with information from a longer temporal range, thereby expanding the effective temporal receptive field of the 3D convolution.

As illustrated in Fig.2(a), given the input feature $I \in R^{N \times T \times C \times H \times W}$, where $N$ denotes the batch size, $T$ denotes the number of frames in temporal dimension, $C$ denotes the channels, and $H, W$ denote the height and width. First, the input is averaged across channels to get $F \in R^{N \times T \times 1 \times H \times W}$. In order to fully fuse the spatial information along the temporal dimension, the feature is reshaped to $F' \in R^{N \times 1 \times T \times H \times W}$ and fed to the 2D convolution $K_1$ with the kernel size of $1 \times 1$, as shown in formula (1).

$$F'' = K_1 * F' \tag{1}$$

$K_1$ takes the time of $F'$ as channel dimension, and each output channel of $F''$ comes from the convolution on all the input frames. Therefore, the relations between different times are established. After the whole-range temporal relation, the effective temporal information excitation can be accomplished by 3D convolution $K_2$, as shown in formula (2).

$$F''' = K_2 * F'' \tag{2}$$

We reshape the output $F'''$ back to $F^* \in R^{N \times T \times 1 \times H \times W}$ and feed it to the Sigmoid activation function to get the single-channel mask. Finally, the temporal relation feature map is produced by the element-wise multiplication of the mask and the input global feature map.

### 2.2.2 MOTION RELATION MODULE

Motion information, which depicts spatio-temporal changes between adjacent frames, can serves as a valuable source for selecting key patches in short range. For moving objects like people or vehicles, their state of motion helps determine the direction and extent of key patch movement. One classical way to model motion information is by using optical flow(Liu & Ma, 2022; Koniusz et al., 2021). However, given the large model size and heavy training costs of Flownet, we propose a lightweight module that provides similar functionality. The motion relation module is built based on the frame differencing method, and relates the motion information at the feature map level by subtracting neighboring frames.

As illustrated in Fig.2(b), given an input $I \in R^{N \times T \times C \times H \times W}$, we first get $M \in R^{N \times T \times 1 \times H \times W}$ by aggregating the information along the channel dimension with $1 \times 1$ 2D convolution. To calculate the changes between frames, the features of different frames are split in the temporal dimension. The feature of the individual frame can be denoted as $M' \in R^{N \times 1 \times H \times W}$ and is fed to $3 \times 3$ 2D convolution $K_3$ for spatial encoding. The encoded feature subtracts from the adjacent feature to excite the motion information at the feature map level.

$$M''_t = K_3 * M'_{t+1} - M'_t \tag{3}$$

$M''_t$ denotes the output motion feature at time $t$. The motion features calculated by formula (3) at different times are concatenated to get the motion mask. The final motion feature map is then generated similarly as in temporal relation.

### 2.3 SPATIAL ADAPTIVE KEY PATCH SELECTION MODULE

Due to the Translation equivariance of CNN, there are spatial correspondences between the feature map and the original image, which means the feature map inherently retains some spatial information. Some prior works select the key patch based on the feature vector(Wang et al., 2021a; 2022b),

in which the valuable spatial information is corrupted. The ability of the feature map to help adaptively select the key patch is demonstrated in this paper. The module provides the coordinates of fractional values without extra weight and permits the gradient back-propagation.

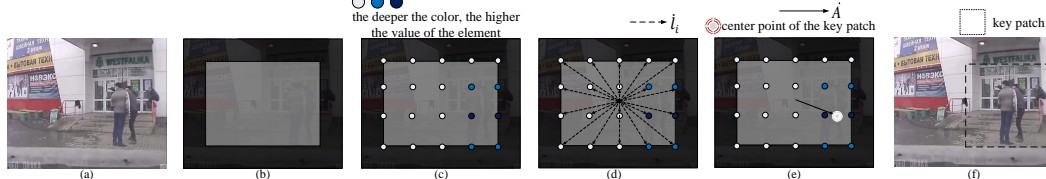

Figure 3: Spatial adaptive key patch selection module.

Given the fixed size and shape(square), the key patch can be determined by its center point. The module outputs the coordinates of the center point and gets the coordinates of other pixels by adding constant offsets. For the input frame in Fig.3(a), to make sure that the patch can be cropped, it is required to leave enough space between the edge of the image and the selected center point. The range of the center point $(x_c, y_c)$ is $(l_a \leq x_c, y_c \leq h_a)$. The area for available center point selection can be presented as the light-color area in Fig.3(b). To fully mine the information in the feature map, $N \times M$ points are taken uniformly from the presented area according to the shape and size of the feature map. Each point corresponds spatially to the element of the feature map, as shown in Fig.3(c)(take the $5 \times 4$ feature map as the example). To build more explicit relations between the taken points and feature map elements, the "shift vectors" that start with the center of the area and end with the taken points are defined, as shown in Fig.3(d). It's obvious that the shift vector $\dot{l}_i$ is a constant vector conditioned only on the row and column of the taken point. And the element $u_i$ is defined as the weight of the shift vector. Finally, the $N \times M$ points are fused according to the information distributed in the feature map, as shown in formula (4).

$$\dot{A} = \sum_{i=1}^{N \times M} u_i \dot{l}_i \tag{4}$$

The end point $(x_t, y_t)$ of $\dot{A}$ is restricted to the available range in Fig.3(b) by $(g(x_t), g(y_t))$ to get the center point of the patch, as shown in Fig.3(e). $g$ can be denoted as formula (5):

$$g(i) = \begin{cases} h_a, i \geq h_a \\ i, l_a < i < h_a \\ l_a, i \leq l_a \end{cases} \tag{5}$$

The pixel of other point $(x_{ij}, y_{ij})$ on the patch can be obtained by adding the offsets to the center point. Since the values of $(x_{ij}, y_{ij})$ are fractional, the pixels with the exact coordinates cannot be directly found in the original frame. We believe that the fractional values can locate the patch and depict the dynamic movements in the video more precisely. Given four points $(\lfloor x_{ij} \rfloor, \lfloor y_{ij} \rfloor), (\lfloor x_{ij} \rfloor + 1, \lfloor y_{ij} \rfloor), (\lfloor x_{ij} \rfloor, \lfloor y_{ij} \rfloor + 1), (\lfloor x_{ij} \rfloor + 1, \lfloor y_{ij} \rfloor + 1)$ ($\lfloor \rfloor$ denotes floor function) surrounding the $(x_{ij}, y_{ij})$ and their pixel values $(s_{ij})_{00}, (s_{ij})_{01}, (s_{ij})_{10}, (s_{ij})_{11}$, we use the interpolation method to get the pixel value $s'_{ij}$ (here we use the bilinear interpolation), as shown in formula (6). The pixel values of other points can be got the same way.

$$s'_{ij} = (s_{ij})_{00} \left( \lfloor x_{ij} \rfloor - x_{ij} + 1 \right) \left( \lfloor y_{ij} \rfloor - y_{ij} + 1 \right) + (s_{ij})_{01} \left( x_{ij} - \lfloor x_{ij} \rfloor \right) \left( \lfloor y_{ij} \rfloor - y_{ij} + 1 \right) \\ + (s_{ij})_{10} \left( \lfloor x_{ij} \rfloor - x_{ij} + 1 \right) \left( y_{ij} - \lfloor y_{ij} \rfloor \right) + (s_{ij})_{11} \left( x_{ij} - \lfloor x_{ij} \rfloor \right) \left( y_{ij} - \lfloor y_{ij} \rfloor \right) \tag{6}$$

The key patch is finally cropped from the input frame, as shown in Fig.3(f). By effectively leveraging the spatial information within the feature map, the selected key patch can capture crucial information for action recognition. The aforementioned process requires no extra weight and few training costs. It can be conveniently used as a plug-and-play module to benefit most of the baselines, and carry out end-to-end training compared to (Wang et al., 2021a). The key patch goes through the key patch extraction network, and the local features are combined with the global features to create the final feature vector.

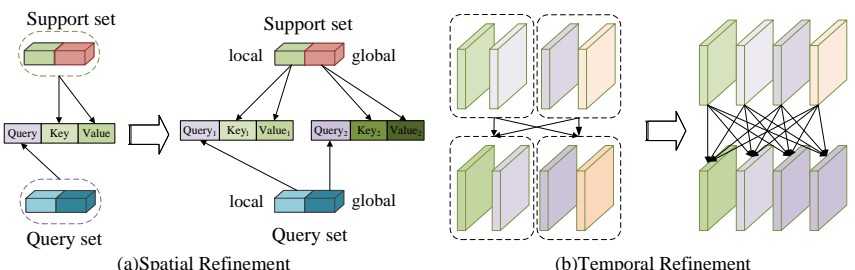

Figure 4: Spatio-temporal refined loss. (a)Spatial refinement. (b)Temporal refinement.

## 2.4 SPATIO-TEMPORAL REFINED LOSS

After extracting the enhanced feature $F_e$, the proper loss function is required to make the model learn the task-aware knowledge. The TRM in (Perrett et al., 2021) uses cross-Transformer attention for action matching but struggles with jointly modeling multi-space features and the constructed tuples may not guarantee the effective learning of frame-level features. To address these problems, we propose the spatio-temporal refined loss, using more spatially and temporally refined feature, as shown in Fig.4. It is the sum of two parts: multi-head attention crosstransformer and temporal refined match loss.

### 2.4.1 SPATIAL REFINEMENT:MULTI-HEAD ATTENTION CROSSTRANSFORMER

Unlike the TRM, which uses a single set of key-value pairs to project input features, our approach incorporates two subspaces for the input features $F_e$: global (the whole frame) and local (the key patch) subspaces. For features in different subspaces, only one single attention function may neglect the differences between them. On the basis of the Crosstransformer, we calculate the loss in the subspace level and adopt the multi-head attention mechanism to make multiple-times projections.

Specifically, the input feature $F_e$ is first divided into two subspaces: the local and global subspaces. Considering the enhanced feature $F_{se}$, the local $F_{sl}$ and global subspace $F_{sg}$ from the support set and the enhanced feature $F_{qe}$, the local $F_{ql}$ and global subspace $F_{qg}$ from the query set, each subspace calculates attention values based on separate set of key-value pairs.

$$
\begin{cases}
S_l = \text{softmax}(\dfrac{F_{ql}K_{sl}^T}{\sqrt{d_{sl}}})V_{sl} \\[3mm]
S_g = \text{softmax}(\dfrac{F_{qg}K_{sg}^T}{\sqrt{d_{sg}}})V_{sg}
\end{cases}
\tag{7}
$$

Where $K$ and $V$ are key-value pairs projected from $F_{sl}$ or $F_{sg}$, $d$ represents the dimension of $K_{sl}$ or $K_{sg}$. Then the two subspaces are aggregated based on multi-head attention to calculate the distance between the samples from the support set and query set.

$$
D_{MH} = cat(S_g, S_l) - F_{qe}
\tag{8}
$$

The distances between samples from the support set and test set can be calculated using formula (8) for action matching. With the integration of the multi-head attention into the loss, the model can automatically attend to the information from different sources. The synergy between the multi-head attention cross-transformer and key patch feature selection enhances both feature representation and matching accuracy simultaneously.

### 2.4.2 TEMPORAL REFINEMENT: FLEXIBLE MATCH LOSS

Considering that sub-sequences of two or three frames can effectively match the action, reinforcing frame-level feature learning can enable the model to learn class-discriminability with different time units, improving the generalization ability. Complex anomalous actions are usually composed of some steps. For instance, a fight may involve pushes, punches, and kicks, but these steps may not follow a fixed sequence in different videos, which means the simple way of matching the steps at the

same time would only work in limited circumstances. Therefore, flexible matching of these steps is required to ensure robustness against misalignment.

As mentioned above, the action consists of multiple frame-level steps, and refined action matching requires comparing frame-level features in the whole temporal range. To match the steps on the different time positions, we use the bidirectional mean Hausdorff Metric(Wang et al., 2022a). For the input frame-level feature sets from the support set $F_s = \{f_s^0, f_s^1, \cdots f_s^i\}$ and query set $F_q = \{f_q^0, f_q^1, \cdots f_q^j\}$, the distance between the two sets $D_{TR}$ is calculated as:

$$D_{TR} = \frac{1}{d} \sum_{f_s^i \in F_s} \left( \min_{f_q^j \in F_q} \left\| f_s^i, f_q^j \right\| \right) + \frac{1}{d} \sum_{f_q^j \in F_q} \left( \min_{f_s^i \in F_s} \left\| f_q^j, f_s^i \right\| \right) \tag{9}$$

Where $\|\|$ means the cosine distance between the features, and $d$ means the dimension of the frame-level feature $f$. The flexible match loss leads to better matching between videos using the classification with multiple time units.

## 3 EXPERIMENTS

**Datasets:**(1)HMDB51(Kuehne et al., 2011) includes 51 actions and 6849 videos. Some actions (walk, run, sit, etc.) are common in real life, and some actions (shoot, kick, punch, etc.) are typically regarded as abnormal. We split this dataset as (Zhang et al., 2020) did, 31,10, and 10 actions are used for training, validation, and testing, respectively.

(2)Kinetics(Carreira & Zisserman, 2017) has 400 actions and 306245 videos in total. We select 100 actions, each of which contains 100 videos, and divide them into training, validation, and test subsets with 64, 12, and 24 actions, just as CMN(Zhu & Yang, 2018) and CMN-J(Zhu & Yang, 2020) do.

(3)UCF-Crime v2(Öztürk & Can, 2021). The UCF Crime v2 dataset is one of the few datasets that includes a variety of anomalous behaviors. We can precisely get the anomaly clips by temporal annotations. Due to the limited number and variety of anomalous behaviors, direct training is not feasible. So we use large normal action data to help the training, and the model is trained on Kinetics and evaluated on UCF-Crime v2.

**Implementation Details:** For a fair comparison with previous methods, we follow the same preprocess steps on the video and uniformly sample 8 frames. These frames are resized to height 256 and augmented with random horizontal flipping and crops.

We trained the models under the 5-way 1-shot and 5-way 5-shot settings using 2×GeForce RTX 3090 Ti, while the model under the 5-way 10-shot setting was trained using 4×GeForce RTX 3090 Ti. To fit in the memory, the query set for each class included 3 videos under the 5-way 1-shot and 5-way 5-shot settings, and 2 videos under the 5-way 10-shot setting.

We use the TRX(Perrett et al., 2021) as the baseline. For the selected patch, the size is fixed to $128 \times 128$. The global and key patch feature extraction networks both use ResNet-50 initialized with Image-Net pretrained weights. The SGD is selected for training the model, and the initial learning rate is set to 0.001. We randomly sample 20000 training episodes for HMDB51 and 30000 training episodes for Kinetics, and report the average accuracy over 10000 random test episodes.

### 3.1 COMPARISON RESULTS

We conduct experiments on the normal action and anomalous action datasets to make a comprehensive comparison of our method versus other state-of-the-art works. Few classical few-shot action recognition methods will evaluate their performance for anomalous actions. So we reimplement these methods and make a fair comparison under the same condition. We show the results on 5-way 1-shot, 5-way 5-shot and 5-way 10-shot benchmarks in Table 1 and Table 2.

According to the results in Table 1 and Table 2, the ATA and OTAM perform better under the 5-way 1-shot setting. But our method can match the action using sub-sequence and be able to reduce some noise, while OTAM and ATA require the whole video. Besides, when the shots increase to 5 or 10, our method has obvious advantages, as shown in Table 1 and Table 2. Our method also gets a noticeable improvement on other methods using the same baseline, namely STRM, SloshNet, and

Table 1: Comparison of our method with others on HMDB and Kinetics. We re-implemented most methods (listed in the bottom half) to obtain more results that are not included in the original papers and to ensure fair comparison. Results marked with * are reported from original papers (listed in the top half). Best results are in bold.

| Method | Publication | HMDB | | | Kinetics | | |
|---|---|---|---|---|---|---|---|
| | | 1-shot | 5-shot | 10-shot | 1-shot | 5-shot | 10-shot |
| ARN(Zhang et al., 2020)* | ECCV 2020 | 45.5 | 60.6 | - | 63.7 | 82.4 | - |
| OTAM(Cao et al., 2020)* | CVPR 2020 | - | - | - | 73.0 | 85.8 | - |
| TRX(Perrett et al., 2021)* | CVPR 2021 | - | 75.6 | - | 63.6 | 85.9 | - |
| ATA(Nguyen et al., 2022)* | ECCV 2022 | 59.6 | 76.9 | - | 74.3 | 87.4 | - |
| STRM(Thatipelli et al., 2022)* | CVPR 2022 | - | 77.3 | - | - | 86.7 | - |
| SloshNet(Xing et al., 2023)* | AAAI 2023 | - | 77.5 | - | - | 87.0 | - |
| BiMACL(Guo et al., 2024)* | ICASSP 2024 | 57.0 | 78.4 | - | 68.1 | 87.6 | - |
| OTAM(Cao et al., 2020) | CVPR 2020 | 49.6 | 60.5 | 62.5 | 68.1 | 79.4 | 80.2 |
| TRX(Perrett et al., 2021) | CVPR 2021 | 51.2 | 73.3 | 78.9 | 63.7 | 84.9 | 88.3 |
| ATA(Nguyen et al., 2022) | ECCV 2022 | **56.5** | 71.5 | 75.9 | **71.1** | 85.0 | 87.8 |
| STRM(Thatipelli et al., 2022) | CVPR 2022 | 50.4 | 73.3 | 79.3 | 66.8 | 85.1 | 88.7 |
| SloshNet(Xing et al., 2023) | AAAI 2023 | 50.9 | 74.0 | 79.1 | 63.0 | 85.8 | 88.7 |
| BiMACL(Guo et al., 2024) | ICASSP 2024 | 53.3 | 73.7 | 77.6 | 64.8 | 85.6 | 88.1 |
| SKPS-Net | | 54.5 | **74.3** | **79.6** | 67.9 | **86.1** | **89.2** |

Table 2: Comparison of our method with others on UCF-Crime v2.

| Method | Publication | UCF-Crime v2 | | |
|---|---|---|---|---|
| | | 1-shot | 5-shot | 10-shot |
| ARN(Zhang et al., 2020)* | ECCV 2020 | - | - | - |
| OTAM(Cao et al., 2020)* | CVPR 2020 | - | - | - |
| TRX(Perrett et al., 2021)* | CVPR 2021 | - | - | - |
| ATA(Nguyen et al., 2022)* | ECCV 2022 | - | - | - |
| STRM(Thatipelli et al., 2022)* | CVPR 2022 | - | - | - |
| SloshNet(Xing et al., 2023)* | AAAI 2023 | - | - | - |
| BiMACL(Guo et al., 2024)* | ICASSP 2024 | - | - | - |
| OTAM(Cao et al., 2020) | CVPR 2020 | **39.3** | 47.9 | 50.6 |
| TRX(Perrett et al., 2021) | CVPR 2021 | 34.4 | 48.2 | 53.2 |
| ATA(Nguyen et al., 2022) | ECCV 2022 | 39.0 | 48.4 | 52.7 |
| STRM(Thatipelli et al., 2022) | CVPR 2022 | 36.5 | 48.7 | 53.3 |
| SloshNet(Xing et al., 2023) | AAAI 2023 | 36.8 | 49.3 | 53.7 |
| BiMACL(Guo et al., 2024) | ICASSP 2024 | 36.6 | 49.6 | 53.5 |
| SKPS-Net | | 37.0 | **50.2** | **54.9** |

BiMACL. It can be concluded that the baseline we utilize is the key factor keeping our method from outperforming the above two methods. From the results of TRX, ATA, and OTAM, we can tell that our baseline TRX has an obvious performance gap under this setting.

Under 5-shot and 10-shot setting, our method achieves the best result on both normal action datasets. The results show that the adaptive selected patch after relation can contain useful contexts and make effective enhancement. This also implies that our method has the potential to benefit most of the action recognition works. For more challenging anomalous action recognition, our method achieved an absolute improvement of 0.6% under the 5-shot setting and 1.2% under the 10-shot setting on the UCF-Crime v2. The results demonstrate that the key patch can highlight critical information and overcome anomalous action recognition problems. Fig.5 shows some visualization results on UCF-Crime v2, the red box locates the key patch. It can be observed that the selected patches can include discriminative objects like the fire and the banner. And because of the relation, the model can attend to the task-relevant patches dynamically with the spatio-temporal changes.

Figure 5: Visualization results of the key patch for robbery, arson, and banner, from top to bottom.

## 3.2 ABLATION STUDIES

To validate the effectiveness of the modules in this paper, we conduct ablation studies under 5-way 1-shot, 5-way 5-shot and 5-way 10-shot settings on the above 3 datasets. The baseline is gradually expanded upon by the addition of proposed modules, as illustrated in Table 3. The spatial adaptive key patch selection module is added first. The results show that the key patches selected at few-computation can emphasize representative information in space, and the extracted key patch feature can make effective feature enhancement. Long-short feature map spatio-temporal relation module then incorporates the spatio-temporal information into the selection, which makes the patch encompass more deep-wise contexts. The spatio-temporal refined loss ensures that the model automatically learns the representative feature.

Table 3: Impacts of the proposed modules in SKPS-Net.

| baseline | spatial adaptive key patch selection | long-short feature map spatio-temporal relation | spatio-temporal refined loss | HMDB | | | Kinetics | | | UCF-Crime v2 | | |
|---|---|---|---|---|---|---|---|---|---|---|---|---|
| | | | | 1-shot | 5-shot | 10-shot | 1-shot | 5-shot | 10-shot | 1-shot | 5-shot | 10-shot |
| ✓ | | | | 51.2 | 73.3 | 78.9 | 63.7 | 84.9 | 88.3 | 34.4 | 48.2 | 53.2 |
| ✓ | ✓ | | | 52.8 | 73.7 | 79.0 | 66.8 | 86.0 | 88.9 | 36.1 | 49.5 | 54.6 |
| ✓ | ✓ | ✓ | | 53.9 | 74.0 | 79.4 | 67.5 | 86.1 | 89.2 | 36.9 | 50.0 | 54.8 |
| ✓ | ✓ | ✓ | ✓ | **54.5** | **74.3** | **79.6** | **67.9** | **86.1** | **89.2** | **37.0** | **50.2** | **54.9** |

Table 4: Impacts of the selected key patch.

| | Kinetics | UCF-Crime v2 |
|---|---|---|
| central cropped patch | 85.5 | 49.4 |
| random cropped patch | 85.6 | 49.4 |
| selected key patch | 86.1 | 50.2 |

The effectiveness of the selected key patch is validated by using two alternative patches:(1) the patch cropped from the center of the image and (2) the patch cropped from the random position of the image. And all the patches have the same size and shape. We compare the results in Table4. The results show that the patch got in naive ways cannot seize the important sign, and the selected patch benefits the recognition by containing helpful context information.

To visually show the performance of the long-short feature map spatio-temporal relation module, we conduct experiments and exhibit the feature maps before and after relation on UCF-Crime v2. Fig.6 shows some examples. Before relation, the global feature map is extracted from the individual frame. And due to the lack of information from other frames, these feature maps occasionally highlight the stationary area, which can be distracting from the action. The feature map after relation takes full advantage of long-range and short-range spatio-temporal information. It can focus on the changing area and suppress the influence of the background area, making the key patch selected more effective.

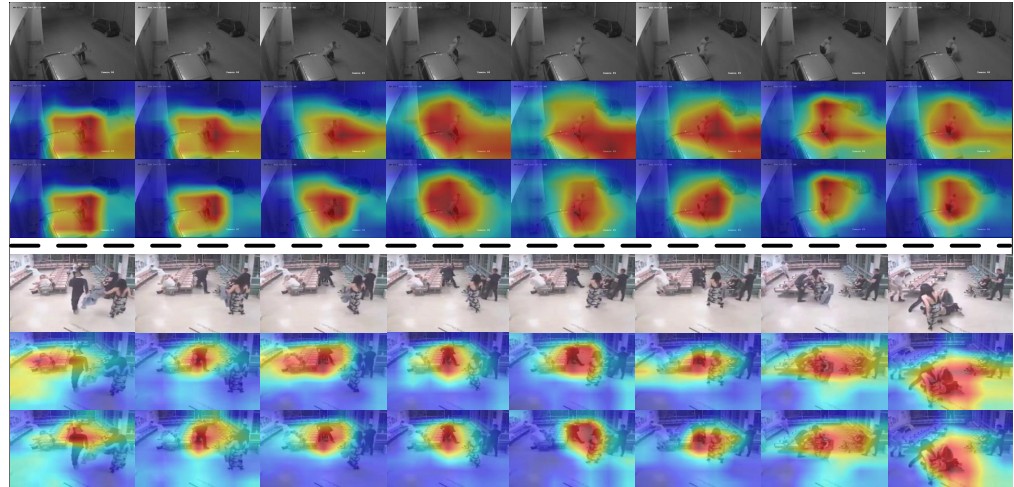

Figure 6: Visualization results of feature maps. We show two examples here. And for each example, the first row shows the input video, and the second and the third rows are the feature maps before and after spatio-temporal relation. Best viewed in color and zoomed in.

## 4 CONCLUSION

The SKPS-Net for anomalous action recognition is proposed in this paper. For more precise modeling of discriminative objects in the local area, the key patch features are extracted to make feature enhancement. In particular, the plug-and-play spatial adaptive key patch selection module locates the informative small-sized area with few extra training costs. Instead of merely adopting the spatial information to extract the key patch feature, the long-short feature map spatio-temporal relation module enriches the long-range temporal and short-range motion information in the selection. Also, the spatio-temporal refined loss is proposed to reinforce effective feature learning on multiple levels. Experiments are conducted on HMDB51, Kinetics, and the anomalous action dataset UCF-Crime v2, showcasing the effectiveness of our SKPS-Net.

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
