# OpenReview forum: "Anomalous Action Recognition via Spatio-temporal Relation and Key Patch Selection"
_ICLR.cc/2025/Conference — ICLR 2025 Conference Withdrawn Submission_

### Official Review · Reviewer_aqbw · 2024-10-29

**Soundness:** 2
**Presentation:** 1
**Contribution:** 2
**Rating:** 3
**Confidence:** 3

**Summary:**

This paper introduces the Spatio-temporal Key Patch Selection Network (SKPS-Net), a new few-shot learning framework designed for anomalous action recognition. To address challenges such as the spatio-temporal and localized nature of anomalous actions, this work propose a spatially adaptive key patch selection module and a long-short feature map spatio-temporal relation module. Additionally, a spatio-temporal refined loss function is introduced to enhance fine-grained feature learning. The experiments demonstrate the advantages of SKPS-Net to some extent.

**Strengths:**

The technical designs appear mostly correct, with the work’s overall structure clear and the methodology practical to follow.

**Weaknesses:**

Presentation：
1. Overall Quality: The tables and figures presented in this work are cluttered, and most figures lack detailed captions, which falls short of the standards expected for high-quality paper—particularly for a top-tier conference.
2. Organization: The introduction provides a vague description of the methods, while the method section appears overly verbose.

Motivation：
The motivations for the three core components seem not to be specifically targeted at anomalous action recognition and could be applicable to normal action recognition as well. This general applicability weakens the paper's persuasiveness.

Technical：
The method’s design comes across as somewhat rough, particularly in the long-short feature map spatiotemporal module, where temporal and motion modeling relies almost on 2D Conv feature extraction. While this approach is feasible, the technical contribution appears limited.

Experiment:
1. While the motivation emphasizes the pursuit of model lightweight, the experiments lack comparisons in terms of parameters or other efficiency metrics.
2. The paper argues that local features are more advantageous than global features for recognition, which is likely correct. However, the patch size is fixed at 128x128. To further verify the effectiveness of local features, it would be beneficial to reduce the patch size or investigate the potential for adaptive adjustments.

**Questions:**

Please refer to the Weaknesses. I'm willing to raise my score if my concerns are well addressed.

---

### Official Review · Reviewer_Kb66 · 2024-10-30

**Soundness:** 2
**Presentation:** 1
**Contribution:** 1
**Rating:** 3
**Confidence:** 4

**Summary:**

The paper introduces a model for action recognition, offering three main contributions: (i) Long-Short feature map spatio-temporal relation, (ii) a Spatial Adaptive Key Patch Selection (SKPS) module, and (iii) new loss functions. Experiments were conducted across various datasets, showing performance improvements.

**Strengths:**

The problem tackled by this paper is relevant and addresses an important area in action recognition.

**Weaknesses:**

1. Long-Short Feature Map Spatio-Temporal Relation:
This component appears to be more of an architectural adjustment tailored to this specific task. The intuition behind combining "long-short" and "spatio-temporal" features is not particularly novel and is widely explored in the literature. Given that similar architectural approaches exist, this element, alongside its modest performance improvements, does not seem to represent a substantial contribution to the field.

2. Adaptive Key Patch Selection (SKPS) Module:
This module could be the most promising aspect of the paper; however, several questions remain. The rationale for the specific modeling approach, particularly for the center vector, is not intuitive. A more straightforward approach, such as adopting bounding box regression—a prevalent method in object detection—might be more effective. Additionally, Table 4, which is central to evaluating this module, shows only marginal gains. More experiments and thorough analyses are necessary to establish the efficacy of this end-to-end salient patch detection module. If this is the paper's primary contribution, pointing to marginal gains in detection accuracy is insufficient.

3. Loss Function:
It is unclear whether this loss function is genuinely effective. In Table 3, its inclusion results in only marginal improvements, possibly within the error margin. While the HMDB 1-shot experiment setting does show some benefit, it’s notable that SKPS-Net underperforms significantly compared to previous works in this setting (Table 1). This casts doubt on the loss function’s value.

4. Writing Quality:
The writing should be improved significantly. The methods section contains overly detailed, almost code-level explanations, which hinder readability. While addressing boundary cases is important in coding, such detail is unnecessary in an academic paper and detracts from clarity. For instance, Section 2.3 reads more like a verbal version of Python code, making it difficult and confusing to follow. Additionally, there is an excessive use of notation, such as the variable "N," which appears with different meanings in L235 and L171. Improving the clarity and flow of the methods section, especially, would enhance the overall quality of the paper.

5. Experimental Results:
Moreover, the main results presented in Table 1 are not particularly strong.

**Questions:**

1. Regarding the Adaptive Key Patch Selection module, what is the input to the module? Is it the raw RGB image or the feature map? Figure 3 suggests the input is RGB, but Figure 1 seems to indicate a combination of raw images and processed feature maps. Additionally, what does the output consist of—a sliced RGB patch, a sliced feature map, or both?

2. In Figure 3(c), what does the blue color represent? The caption refers to it as "the value," but this needs clarification. What specifically does this value indicate?

3. For my understanding, is "u" produced per image since it’s used to select the key patch?  is "u" implemented as a learnable parameter (e.g., nn.Parameter)?

---

### Official Review · Reviewer_JXW8 · 2024-11-02

**Soundness:** 2
**Presentation:** 1
**Contribution:** 1
**Rating:** 3
**Confidence:** 3

**Summary:**

This paper introduces a Spatio-Temporal Key Patch Selection Network (SKPS-Net) for the anomalous action detection, which mainly presents a spatial adaptive key patch selection module, a long-short feature map spatio-temporal relation module, and a spatio-temporal refined loss.

**Strengths:**

The motivation of this paper is clear.

**Weaknesses:**

The SKPS-Net is proposed to detect anomalous actions in this paper, but the comparative models belong to the few-shot action recognition methods in the experiments.

**Questions:**

1. This paper focuses on the anomalous action recognition, but many existing few-shot action recognition methods are introduced and the related anomalous action recognition models are not mentioned, such as CLAWS, CLIP-TSA, DGGAN, and so on. Likewise，the proposed SKPS-Net is only compared with some few-shot action recognition methods, but not with the anomalous action recognition methods in the experiments.
2. In the spatial adaptive key patch selection module, is the key patch selected according to the feature map? If yes, why is the original input image fed into the Spatial Adaptive Key Patch Selection module in Figure 1?
3. In Section 2.4.1, authors do not clearly explain how the support set and query set are defined.
4. Authors only use the UCF-Crime v2 dataset to evaluate the performance of the proposed SKPS-Net for the anomaly behavior detection. The experimental results can't fully validate the effectiveness of the proposed method. There are some commonly used datasets for the anomaly behavior detection, such as Shanghai Tech Campus, XD-Violence.

---

### Official Review · Reviewer_vPQ7 · 2024-11-04

**Soundness:** 1
**Presentation:** 2
**Contribution:** 1
**Rating:** 3
**Confidence:** 4

**Summary:**

This paper proposes a SKPS-Net for anomalous action recognition. The SKPS-Net is built based on TRX, and includes a spatial adaptive key patch selection module for highlighting the local information, with a long-short feature map spatio-temporal relation module to enrich long and short-range temporal information. The anomalous action recognition is formulated as a few-shot learning task, owing to the lack of data and the need to apply models trained in large normal action data towards anomalous action data.

**Strengths:**

The authors discussed on anomalous action recognition which is an important task to enable video analysis modules to be applied to security applications. The formulation of such task as a few-shot learning task is indeed reasonable given the limited available training data with the availability of large-scale normal video data. The empirical results are in general descent, with the visualization results provided for the key patch and the feature map with activation (Class Activation Map, CAM).

**Weaknesses:**

Despite some improvements made in the performance on some action recognition datasets, there are serious concerns over the novelty of the method and the clarity in the presentation. The authors may consider addressing the following concerns:

1. The idea of patch-based action recognition is not novel. Though may be formulated differently, papers dating back to 2021 [a] and 2022 [b] have already leveraged on key points or patches for action recognition. However, none of them are mentioned, reviewed or compared.
2. Why is the method still reliant on CNN when Transformers have already been the mainstream of action recognition? Even if the authors argue that CNNs may be less resource intensive than Transformers, there have been several works [c-e] where previous researchers are able to greatly improve the efficiency of Transformers. Nonetheless, Transformers are replacing CNNs for a reason, with its great ability to include both short-term and long-term spatial dependencies, which is mentioned even at the introduction of self-attention modules for action recognition [f].
3. Even for using CNNs, it should be noted that most action recognition papers leverage on 3D CNNs, which does not, as the author claims on Page 3 Line 151, "only reserves the spatial information from the individual frames". This statement only applies to pure 2D CNN-based action recognition, which is already out-dated in the research field even 3-4 years ago. Even the authors do not purely rely on 2D CNNs.
4. It should also be noted that papers such as [g] also attempts to obtain relational features from CNN maps, which is also not included in the authors' review or comparison.
5. The author states that the spatial adaptive key patch selection module "permits the gradient back-propagation". However, I do not see how the back-propagation is permitted, with the non-continuous floor function and various piecewise functions applied. Could the authors ellaborate on how the back-propagation gradient is computed?
6. Details of the implementation are not clarified, including:
- a. For UCF-Crime v2, how many classes are used for training, validation, and testing?
- b. For UCF-Crime v2, in total how many videos are there?
- c. For HMDB51 and Kinetics, how are the classes for training, validation, and testing selected?
- d. For fair comparison, are all methods re-run with TRX as the common baseline?
7. Lastly, it is observed the the SKPS-Net does not perform well on all 1-shot settings, why does the SKPS-Net fall short on this setting?
8. Also, the code is currently NOT available, which means that the reproducibility of the result is not verified.

[a] Cao, H., Xu, Y., Yang, J., Mao, K., Yin, J., & See, S. (2021). Effective action recognition with embedded key point shifts. Pattern Recognition, 120, 108172.
[b] Xiang, W., Li, C., Wang, B., Wei, X., Hua, X. S., & Zhang, L. (2022, October). Spatiotemporal self-attention modeling with temporal patch shift for action recognition. In European Conference on Computer Vision (pp. 627-644). Cham: Springer Nature Switzerland.
[c] Weng, Y., Pan, Z., Han, M., Chang, X., & Zhuang, B. (2022, October). An efficient spatio-temporal pyramid transformer for action detection. In European Conference on Computer Vision (pp. 358-375). Cham: Springer Nature Switzerland.
[d] Yuan, K., Yu, Z., Liu, X., Xie, W., Yue, H., & Yang, J. (2025). Auformer: Vision transformers are parameter-efficient facial action unit detectors. In European Conference on Computer Vision (pp. 427-445). Springer, Cham.
[e] Xu, Y., Cao, H., Yang, J., Mao, K., Yin, J., & See, S. (2021). PNL: Efficient long-range dependencies extraction with pyramid non-local module for action recognition. Neurocomputing, 447, 282-293.
[f] Wang, X., Girshick, R., Gupta, A., & He, K. (2018). Non-local neural networks. In Proceedings of the IEEE conference on computer vision and pattern recognition (pp. 7794-7803).
[g] Xu, Y., Yang, J., Mao, K., Yin, J., & See, S. (2021). Exploiting inter-frame regional correlation for efficient action recognition. Expert Systems with Applications, 178, 114829.

**Questions:**

Please see the Weaknesses section for details.

---

### Note · Authors · 2024-11-13

I have read and agree with the venue's withdrawal policy on behalf of myself and my co-authors.